# Advances in Gut Microbiota-Targeted Therapeutics for Metabolic Syndrome

**DOI:** 10.3390/microorganisms12050851

**Published:** 2024-04-24

**Authors:** Yu Gao, Wujuan Li, Xiaoyu Huang, Yuhong Lyu, Changwu Yue

**Affiliations:** 1Yan’an Key Laboratory of Microbial Drug Innovation and Transformation, School of Basic Medicine, Yan’an University, Yan’an 716000, China; gy944013192@163.com (Y.G.); lwj2117101041@163.com (W.L.); suiyu21@outlook.com (X.H.); yuhonglyu@126.com (Y.L.); 2Shaanxi Engineering and Technological Research Center for Conversation and Utilization of Regional Biological Resources, Yan’an University, Yan’an 716000, China

**Keywords:** gut microbiome, metabolic syndrome, probiotics, prebiotics, dysbiosis

## Abstract

Previous investigations have illuminated the significant association between the gut microbiome and a broad spectrum of health conditions, including obesity, diabetes, cardiovascular diseases, and psychiatric disorders. Evidence from certain studies suggests that dysbiosis of the gut microbiota may play a role in the etiology of obesity and diabetes. Moreover, it is acknowledged that dietary habits, pharmacological interventions, psychological stress, and other exogenous factors can substantially influence the gut microbial composition. For instance, a diet rich in fiber has been demonstrated to increase the population of beneficial bacteria, whereas the consumption of antibiotics can reduce these advantageous microbial communities. In light of the established correlation between the gut microbiome and various pathologies, strategically altering the gut microbial profile represents an emerging therapeutic approach. This can be accomplished through the administration of probiotics or prebiotics, which aim to refine the gut microbiota and, consequently, mitigate the manifestations of associated diseases. The present manuscript evaluates the recent literature on the relationship between gut microbiota and metabolic syndrome published over the past three years and anticipates future directions in this evolving field.

## 1. Introduction

The intestinal microbiota, often termed the gut microbiome, is composed of trillions of microorganisms that inhabit the gastrointestinal tract. These microbes, which encompass bacteria, fungi, viruses, and other forms of life, engage in a critical relationship with human health. The dense populations of bacteria residing within our gut tissues can potentially pose health risks such as inflammation and infection. As a result, the immune system has evolved to maintain the symbiotic balance between the host and its microbiota. Inversely, the gut microbiota frequently exerts an immunomodulatory effect that is vital for sustaining host immune homeostasis. In recent years, advancements in metagenomics, 16S rRNA sequencing, and other molecular techniques have facilitated significant progress in the study of intestinal microbiota. Current research methodologies include metagenomic sequencing, 16S rRNA gene sequencing, metatranscriptomics, and metabolomics analysis. These approaches allow for the elucidation of correlations between various diseases and the gut microbiome, providing a theoretical foundation for the prevention, diagnosis, and treatment of these conditions [1]. Metagenomic sequencing provides a comprehensive analytical approach for examining microbial communities, revealing the complete gut microbiome profile, including species identification and their relative abundances [2]. 16S rRNA gene sequencing is employed as a targeted method for bacterial analysis, contributing to an understanding of bacterial diversity and population estimations. Metatranscriptomic and metabolomic analyses offer perspectives into microbial activity states and their impact on the host’s physiological processes. Metabolic syndrome (MS) is a complex metabolic disorder hallmarked by insulin resistance, abdominal obesity, hypertension, hyperglycemia, and dyslipidemia [3]. Insulin resistance serves as the central feature of MS, reducing the body’s sensitivity to insulin and leading to glucose metabolism dysfunction. Abdominal obesity is recognized as a significant contributor to insulin resistance, with inflammatory factors and hormones released by adipose tissue exacerbating this condition [4]. Hypertension is closely associated with insulin resistance, affecting vascular endothelial cells and causing vascular dysfunction. Hyperglycemia results from a combination of insulin resistance and insufficient insulin secretion, which can lead to vascular damage, neuropathy, and nephropathy if persistent. Dyslipidemia involves elevated triglycerides, increased LDL-C, and decreased HDL-C levels, directly predisposing individuals to atherosclerosis and elevating the risk of cardiovascular events.

Emerging scientific advancements and refined research methodologies have driven significant progress in the study of metabolic diseases [5]. Perturbations in the gut microbiota may affect insulin secretion and action, thereby impacting glycemic control. Modulating the gut microbiome thus emerges as a novel therapeutic strategy for diabetes treatment. Adipokines play a crucial role in the onset and progression of obesity. The gut microbiota may contribute to the development of obesity by modulating adipokine expression and secretion [6]. It has been demonstrated that intestinal microbes can mitigate hypertension through various mechanisms, including altering the gut microbial composition and producing short-chain fatty acids [7]. Gut microbes influence lipid levels through various mechanisms, including fat absorption and metabolism, as well as cholesterol synthesis and excretion. Over the past three years, notable progress has been made in metabolic disease research, particularly concerning diabetes, obesity, and hypertension. These advancements have not only yielded innovative therapeutic strategies and methods but also presented fresh perspectives and directions for future inquiry. Figure 1 shows the metabolic syndrome caused by obese mice.

## 2. Gut Microbiota and Metabolic Syndrome

Over the past three years, the intricate relationship between metabolic diseases and the gut microbiome has attracted considerable attention within the scientific community. While significant advancements have been achieved, numerous facets still require further investigation. For example, there is an urgent need to enhance our comprehension of the gut microbial community’s structure and function, as well as its role in the development and progression of metabolic disorders. Moreover, there is a crucial necessity to devise more effective strategies for modulating the gut microbiome to prevent and treat these conditions. Lifestyle factors, obesity, and the gut microbiome are recognized as pivotal contributors to the risk of metabolic dysregulation [8]. A variety of gut microbes, including Firmicutes, Enterobacter, Bacteroidetes, Lactobacillus rhamnosus PL60, Escherichia coli, Staphylococcus aureus, and Bifidobacterium, have been implicated in conditions such as obesity and diabetes. These microorganisms can influence host metabolism through their metabolites. For instance, the interaction between bile acids and the gut microbiota can alter bile acid composition, which subsequently modulates host metabolic pathways via receptors such as TGR5 and FXR signaling [9]. Alterations in the gut microbiota composition are evident in the development of metabolic syndrome and obesity [10]. The transformation of the gut microbiome is associated with the emergence and progression of diabetes, and it can impact glucose levels through multiple mechanisms. Investigations have elucidated that the gut microbial community structure in diabetic patients significantly deviates from that of healthy individuals. Analogously, perturbations in the intestinal microbial community are implicated in the onset and progression of obesity, with the community structure in obese patients diverging substantially from that of their healthy peers. Moreover, shifts in the intestinal microbiome are linked to the development of hypertension and can influence blood pressure through various pathways [11]. A multitude of studies have underscored the gut microbiota’s involvement in diverse aspects of metabolic syndrome, including insulin resistance, dyslipidemia, atherosclerosis, hepatic steatosis, and elevated blood pressure [12]. Figure 2 shows the metabolism of gut microbiota and its impact on target organs.

### 2.1. Gut Microbiota and Obesity

Obesity represents a significant global health challenge, having attained epidemic proportions. In recent years, the role of the gut microbiome in obesity has attracted considerable scrutiny [11]. While the fundamental cause of obesity is an imbalance between energy intake and expenditure, disparities in the gut microbial ecology between healthy individuals and those who are obese may exert an influence on energy homeostasis. In essence, individuals with a propensity for obesity may harbor specific gut microbial communities that enable more efficient energy extraction and/or storage from a given diet. Traditionally, obesity has been characterized as excessive adiposity that is detrimental to health and has been clinically evaluated using the body mass index (BMI), which is determined by dividing weight (in kilograms) by the square of height (in meters) [13]. A BMI of 25 or higher is categorized as overweight, while obesity is defined as having a BMI of 30.0 or above [14]. Dietary interventions have the potential to alleviate obesity. Meslier and colleagues elucidated that a Mediterranean diet augmented the presence of microbial genes associated with carbohydrate degradation and butyrate metabolism in fecal bacteria, thereby ameliorating the health status of obese subjects [15]. Ma and coworkers emphasized that a compromised gut barrier function could result in augmented permeability, enabling microbiota-derived endotoxins, such as lipopolysaccharide (LPS), to infiltrate systemic circulation. Toll-like receptor 4 (TLR4) detects LPS, activating proinflammatory signaling pathways that induce insulin resistance and exacerbate obesity. Nevertheless, dietary spermidine has been observed to safeguard gut barrier function and curtail permeability. Moreover, spermidine has manifested anti-obesity effects in DIO mice.

Traditional Chinese medicine has been recognized for its role in managing obesity [16]. Xu and colleagues reported that panax notoginseng saponins (PNS) regulated the intestinal microbiota in diet-induced obese (DIO) mice, promoting brown adipose tissue (BAT) thermogenesis and beige adipocyte biogenesis through the leptin-AMPK/STAT3 pathway, thereby increasing energy expenditure and alleviating obesity [17]. A prevalent strategy involves modulating the gut microbiota with probiotics and prebiotics to mitigate obesity. Kong and coworkers conducted a randomized, double-blind, placebo-controlled trial on Prader–Willi syndrome (PWS) patients, revealing that Lactobacillus reuteri advantageously modified the gut microbiota, assisting weight loss and enhancing gut health [18]. Allegretti and colleagues reviewed mouse studies suggesting that the gut microbiota influences obesity by altering anorexigenic hormones such as glucagon-like peptide 1 (GLP1) and bile acids, impacting lipid metabolism Their double-blind study demonstrated that fecal microbiota transplantation (FMT) capsules obtained from lean donors were well-tolerated and induced lasting changes in the gut microbiome and bile acid profile akin to those of lean donors, ameliorating obesity-related health concerns [19]. Solito and colleagues observed that treatment with Bifidobacterium breve strains BR8 and B03 enhanced insulin sensitivity in obese young adults. Exercise training has been demonstrated to enhance the gut microbiome and diminish endotoxemia [20]. The gut microbiota is believed to foster systemic low-grade inflammation and insulin resistance through the discharge of endotoxins, particularly lipopolysaccharide (LPS). Quiroga and colleagues discovered that exercise training substantially decreased Proteobacteria and gamma bacteria, resulting in a lesser degree of obesity compared to control groups [21]. It is widely acknowledged that exercise training can mitigate the effects of obesity. Research conducted by Quiroga et al. has demonstrated that a reduction in the phylum Proteobacteria and gamma bacteria significantly curtails the activation of NLRP inflammatory signaling pathways associated with obesity, thereby contributing to the alleviation of this condition [22]. Additionally, Sbierski King et al. have elucidated that calorie restriction diminishes the prevalence of microbiota linked to obesity, which are also implicated in systemic inflammation, carcinogenesis, and metabolic disorders, subsequently ameliorating immune senescence and subduing low-grade inflammation [23].

### 2.2. Gut Microbiota and Type 2 Diabetes Mellitus

Diabetes mellitus (DM) is characterized by deficient insulin secretion, action, or both, culminating in hyperglycemia [24]. The prevalence of type 2 diabetes mellitus (T2D) is on the rise globally, with forecasts predicting that approximately 642.1 million individuals will be affected by the disease by the year 2040 [25]. The emergence of drug resistance in T2D has prompted interest in modulating the gut microbial composition to reestablish a healthful host–microbiota relationship as a crucial approach for improving T2D. The gut microbiome significantly impacts systemic metabolism and represents a pervasive therapeutic target and pathway for managing type 2 diabetes. Diabetic complications are a leading cause of death among diabetic patients. In recent years, substantial progress has been achieved in research related to diabetic complications, including nephropathy and retinopathy. A multicenter randomized double-blind controlled clinical trial conducted by Zhang and colleagues involving 409 newly diagnosed T2D patients corroborated the hypoglycemic effect of the bacteriostatic compound berberine (BBR) in a Chinese cohort [26]. Further investigations revealed that the synergistic use of berberine with probiotic formulations, such as Prob + BBR, could substantially mitigate postprandial dyslipidemia—a contributory factor to cardiovascular diseases in T2D. Additionally, Bifidobacterium breve has emerged as a potential effective ingredient in probiotic BBR formulations to improve lipid profiles, with evidence suggesting a synergistic effect between Bifidobacterium-containing probiotics and BBR in reducing postprandial lipidemia [27]. Moreover, experimental evaluations by Perraudeau and coworkers indicated that a novel probiotic formulation, WBF-011, supplemented with diverse strains, enhanced postprandial glycemic control [28]. Adeshirlarijaney compiled an array of plant-derived products, including berberine, resveratrol, alliin, capsaicin, betaine, anthocyanins, and cranberry proanthocyanidins, which exhibit potential biological activities and antidiabetic effects, possibly through the modulation of the gut microbiome [29]. Flaxseed oil (FO), rich in plant-derived omega-3 polyunsaturated fatty acids (PUFAs) such as α-linolenic acid (ALA), has demonstrated benefits in chronic metabolic diseases. Dietary intake of FO has been shown to ameliorate T2DM [30] by inhibiting inflammation and modulating the gut microbiota in Sprague Dawley rat models. Patients with T2DM frequently exhibit fecal microbiota dysbiosis [31]. Chen and colleagues demonstrated experimentally that Simiao pills could alleviate intestinal microbiota dysbiosis in diabetic patients. Furthermore, the beneficial effects of SMW on insulin resistance and hepatic lipid accumulation in mice fed a high-fat diet (HFD) were partially mediated by modulating bile acid profiles and gut microbiota composition. A randomized clinical study by Chen and coworkers posited that a high-fiber diet could improve glucose homeostasis, serum metabolome, and systemic inflammation in T2D subjects [9]. An increased abundance of Lactobacillus, Bifidobacterium, and Akkermansia suggested that a high-fiber diet augmented the proportion of beneficial gut microbes while decreasing the presence of opportunistic pathogens such as Desulfovibrio and Klebsiella. Traditional Chinese medicine (TCM) [31], a prominent complementary and alternative medicine, has been found to regulate intestinal flora and enhance glucose metabolism in T2D patients. In a double-blind randomized placebo-controlled trial [32], repeated fecal microbiota transplantation (FMT) enhanced the level and duration of microbiota engraftment in obese T2D patients, with lifestyle interventions combined with FMT resulting in more favorable microbiota alterations in recipients [33]. A randomized controlled trial by Birkeland and colleagues indicated that daily supplementation with inulin-based fructans induced moderate yet significant increases in fecal bifidobacteria, total short-chain fatty acids (SCFAs), acetate, and propionate levels in patients with T2D. The study noted that overall microbial diversity or fecal butyrate levels were unaffected; however, these prebiotic fibers hold moderate potential for improving the gut microenvironment in T2D [34].

### 2.3. Gut Microbiota and Hypertension

The prevalence of hypertension in China is remarkably high and continues to rise. Hypertension is known to aggravate the condition of atherosclerosis, which in turn leads to an increased incidence of cardiovascular diseases [35]. The cost of therapeutic medications for hypertension can be prohibitive, and issues with resistance may reduce their effectiveness in certain patient populations. Elevated blood pressure significantly increases the risk of cardiovascular disease and premature mortality. As such, hypertension is considered one of the most significant public health concerns in China [36]. Over the past decade, a substantial body of evidence has emerged to support the role of the gut microbiome in regulating blood pressure. In the last five years, research has shifted from establishing associations to proving causation, with studies utilizing germ-free animals, antibiotic treatments, and the direct supplementation of microbial metabolites [37]. There is a growing consensus that probiotics may offer an alternative nonpharmacological approach to reducing blood pressure [38]. Notably, a study by Fan and colleagues conducted a multicenter randomized placebo-controlled double-blind trial using fecal microbiota transplantation (FMT) to investigate the therapeutic potential of gut microbiota interventions in essential hypertension [39]. Richards and coworkers have demonstrated that hypertension induces dysbiosis of the gut microbiota, which can, in turn, perpetuate hypertension, suggesting a bidirectional causality [40]. The modification of the gastrointestinal microbiome as a treatment for hypertension is under increasing scrutiny, with Xia and colleagues revealing that exercise not only lowered blood pressure but also altered the gut microbiome. Exercise-induced sustained improvements in the microbiome–gut–brain axis result in lasting reductions in blood pressure and the amelioration of some hypertensive lesions [41]. Xue and colleagues concluded that an adequate intake of dietary fiber is an effective strategy for improving blood pressure among individuals with hypertension or prehypertension [36]. Grape pomace (GP), a by-product of wine production, is rich in polyphenols and dietary fiber. It has been demonstrated that the polyphenols and dietary fiber present in grape pomace can favorably influence the gut microbiota. Grape pomace has the potential to increase the population of beneficial bacteria, such as Bifidobacterium and lactic acid bacteria, while concurrently reducing the number of pathogenic bacteria. This modulation may enhance gut health, which could positively impact cardiac health and potentially regulate high blood pressure. Some investigations suggest that soy-based foods may protect against hypertension by modulating the gut microbiome [42]. However, some individuals may experience adverse reactions to soy, potentially linked to their specific gut microbiome. Hence, dietary soy intake could shape the microbiome by suppressing particular taxa and prevent hypertension exclusively in individuals with a soy-responsive microbiome. Lv and colleagues corroborated these findings in a clinical trial involving hypertensive women and men from a northwest Chinese population, providing evidence of fecal gut microbiome signatures [43]. These results further support the hypothesis that dysbiosis of the gut microbiota might be implicated in the pathogenesis of hypertension [44]. Moreover, reducing dietary sodium has been shown to elevate circulating short-chain fatty acids (SCFAs), which can affect blood pressure and arterial compliance through alterations in SCFA levels [45].

### 2.4. Gut Microbiota and Hyperlipidemia

Hyperlipidemia, characterized by elevated blood levels of lipids such as cholesterol and triglycerides, has seen an increasing incidence in recent years due to improved living standards and dietary habits [46]. This prevalent metabolic disorder is responsible for over 17 million deaths annually worldwide through cardiovascular and cerebrovascular diseases, with coronary heart disease and stroke being the leading causes of death. In China, the prevalence of hyperlipidemia continues to rise, currently affecting over 100 million individuals, with a higher prevalence among men than women and an increase with age [47]. The significance of hyperlipidemia as a metabolic disease is undeniable, and recent studies have highlighted the close relationship between gut microbiota and its onset and progression. Intestinal microbes modulate the host’s lipid metabolism via various mechanisms, including the regulation of the gut flora balance and the production of beneficial metabolites. Further research is warranted to elucidate the mechanisms and therapeutic potential of the gut microbiome in combating hyperlipidemia. Fecal microbiota transplantation experiments have unveiled the pathogenic role of gut microbiota in hyperlipidemia, and the regulatory functions of microbiota-derived metabolites such as bile acids, lipopolysaccharides, and short-chain fatty acids have been partially revealed. Interventions targeting the gut microbiota, including prebiotics, probiotics, fecal microbiota transplantation, and natural herbs, have demonstrated efficacy in managing hyperlipidemia. These treatments may influence host lipid metabolism by enhancing the composition and function of the gut microbiota, thereby reducing lipid levels and cardiovascular risks [48]. Xu and colleagues, in a randomized controlled trial, reported that oat consumption significantly lowered total cholesterol (TC) and low-density lipoprotein cholesterol (LDL-C), exerting a prebiotic effect on the gut microbiome. Phytochemicals in oats, such as β-glucans and β-glucanase inhibitors, are suggested to play a role in lipid reduction. Studies have shown that certain bacteria can produce cholesterol, while others can alter cholesterol absorption and metabolism [49]. For instance, Akkermansia, found in reduced numbers in individuals with hyperlipidemia, can modulate blood lipids by balancing the gut microbiota and producing beneficial metabolites. Zhang and colleagues observed changes in fecal Bifidobacterium breve levels with combined probiotic and berberine therapy, indicating a potential synergistic effect on the gut microbiome [27]. Probiotics have also been linked to reduced serum cholesterol levels and improved lipid profiles, emphasizing their preventive and therapeutic value against hyperlipidemia [50].

Butyrate, a key short-chain fatty acid produced by gut microbial fermentation, has been shown to regulate serum cholesterol levels favorably. Lim and colleagues investigated the benefits of a novel oil mixture on lipid profiles by stimulating butyrate production [51]. Advances in gene sequencing technology have deepened our understanding of the gut microbiome’s structure and function, providing new tools for exploring its relationship with hyperlipidemia. Dietary modifications and probiotics have been clinically proven to effectively reduce blood lipid levels, and differences in gut microbiota composition have been noted between healthy individuals and those with hyperlipidemia, corroborating the link between gut microbiota and lipid disorders [52]. Furthermore, gut microbes can influence lipid synthesis and catabolism by modulating host metabolic pathways, offering a foundation for developing novel therapeutic strategies [53]. Over the past three years, significant attention has been given to the gut microbiota’s role in hyperlipidemia, yet many questions remain to be addressed. A deeper comprehension of the gut microbial community’s structure and function and its impact on lipid metabolism is necessary. Additionally, there is a need to develop more effective strategies for modulating the gut microbiome to mitigate hyperlipidemia risks. Table 1 shows clinical trials and animal studies on the treatment of metabolic syndrome with biotherapy.

## 3. Pharmacological Modulation of Gut Microbiota as a Therapeutic Approach for Metabolic Syndrome

### 3.1. Probiotics, Prebiotics, Metabiotics, and Synbiotics

Since the year 2020, there has been a progressive surge in scholarly articles pertaining to the roles of probiotics, prebiotics, metabiotics, and synbiotics within metabolic syndrome treatment, primarily focusing on the impact these microbial agents have on metabolic syndrome and their potential mechanisms of action. It has been indicated that probiotics can effectively redress the intestinal dysbiosis observed in metabolic syndrome patients. It has been discovered that the intestinal milieu can be ameliorated through the administration of probiotics such as Bifidobacterium and lactic acid bacteria, which serve to augment the population of salubrious gut bacteria while curtailing the proliferation of deleterious bacteria [57]. This amelioration contributes to reduced levels of blood glucose, lipids, and blood pressure, dampened inflammatory responses, and enhanced insulin sensitivity.

Prebiotics, classified as soluble dietary fibers, have also demonstrated therapeutic utility in managing metabolic syndrome. Prebiotics foster the proliferation of beneficial gut bacteria and inhibit the propagation of harmful bacteria, thereby upholding the equilibrium of the intestinal microbiota [11]. Investigations have revealed that the consumption of prebiotics can mitigate blood glucose, lipid, and blood pressure levels, as well as ameliorate insulin resistance and inflammation [58]. Furthermore, metabiotics, postulated as processed derivatives of probiotics, have exhibited therapeutic potential in addressing metabolic syndrome. Metabiotics exhibit superior resistance to acidity and stability, enabling them to endure and exert influence within the intestinal tract. Studies have shown that metabiotics can modulate the configuration of the gut microbiome, elevating the presence of beneficial bacteria and curtailing the expansion of injurious bacteria, thus enhancing metabolic functions. Synbiotics, a composite formulation integrating probiotics and prebiotics, have been employed in metabolic syndrome treatment regimens. Synbiotics operate via multifaceted mechanisms, including the regulation of intestinal flora homeostasis, the augmentation of immune system functionality, and the diminution of inflammatory responses. Empirical evidence suggests that synbiotics significantly lower blood glucose, lipid, and blood pressure levels and ameliorate symptoms associated with insulin resistance and obesity.

### 3.2. Fecal Microbiota Transplantation

Fecal microbial transplantation (FMT), also referred to as intestinal microbiota transplantation, constitutes a therapeutic intervention wherein fecal material from a healthy donor is administered to a recipient for the purpose of modulating the intestinal microbiota. This procedure has garnered increasing attention for its potential to directly reconstitute the gut microbiota, offering prospects for the management of metabolic syndrome. However, the safety and efficacy profile of FMT warrants comprehensive investigation to validate its clinical utility. An expanding corpus of research has been dedicated to exploring the capacity of FMT to enhance insulin sensitivity in individuals afflicted with obesity and metabolic syndrome. Such studies typically involve the transfer of fecal microbiota from individuals exhibiting a lean phenotype to those suffering from obesity or metabolic syndrome, followed by an assessment of the impact on insulin responsiveness [59]. FMT has been applied across a broad range of conditions, thereby demonstrating its adaptability in addressing a diverse array of pathologies [19,60]. In adolescent populations, the transfer of fecal microbiome has exhibited the potential to mitigate obesity and improve metabolic parameters. Ng and colleagues reported that iterative FMT procedures in obese patients with type 2 diabetes mellitus (T2DM) significantly enhanced the engraftment rate and persistence of the transferred microbiota [61]. When combined with lifestyle modifications, FMT elicited more pronounced alterations in the gut microbiome, resulting in improved lipid profiles and reduced liver stiffness among the study participants [33].

### 3.3. Others

Intermittent fasting (IF) is characterized as a dietary regimen that involves the scheduled restriction of eating periods, also known as time-restricted feeding. This approach has been observed to exert a significant influence on the composition of the gut microbiota, which may subsequently contribute to the enhancement of cardiovascular health [56]. IF has been documented to induce notable alterations within the gut microbial community. For instance, research has demonstrated that IF can increase the production of short-chain fatty acids (SCFAs), a class of gut microbial metabolites that confer an array of health benefits to the host, such as improved insulin sensitivity, reduced inflammatory responses, and the suppression of detrimental bacteria. Lipopolysaccharides, integral components of the bacterial cell wall, have the potential to initiate inflammation. By reducing circulating levels of LPS, IF may contribute to the alleviation of cardiovascular risks associated with chronic inflammation. Studies have additionally revealed that IF can lower blood pressure, ameliorate lipid profiles, and reduce body weight—all recognized as risk factors for cardiovascular disease. Moreover, IF may induce specific genetic alterations in the gut microbiota related to carbohydrate metabolism, suggesting that modifications in dietary patterns could potentially modulate host metabolism and health by affecting the gene expression of gut microbes.

Dietary fiber derived from whole grains, vegetables, and fruits serves as a nutrient source for the gut microbiota and can significantly influence the composition, diversity, and richness of the microbiome [62]. This effect is attributed to dietary fiber providing a variety of substrates for particular microbial species equipped with the enzymes necessary for degrading these complex carbohydrates. For example, soluble fiber can be fermented by certain gut bacteria to produce beneficial compounds such as SCFAs. Dietary supplements, including probiotic and prebiotic formulations, offer a convenient method for supplementing the nutrients essential for intestinal flora [63]. Investigations have demonstrated that purified citrus polymethoxy-flavonoid-rich extract (PMFE) effectively mitigates metabolic syndrome (MetS) induced by a high-fat diet (HFD), and it attenuates the dysbiosis of the gut microbiota. It has been indicated that the metabolic protective effects conferred by PMFE are contingent upon the presence of gut microbiota [64].

## 4. Chrononutrition and Metabolic Syndrome

Regular feeding patterns have been shown to entrain the peripheral circadian clock, while the peripheral clock systems are known to govern the absorption, distribution, metabolism, and excretion of nutrients. This suggests a reciprocal interaction between circadian clocks and nutrition/food intake, a phenomenon termed “chrononutrition” [65]. A study by Lujan Barroso L et al. examined 3644 participants from the European Cancer Prospective Survey and the Spanish Nutrition Study. The findings indicated that among Spanish adults, a higher volume of breakfast correlates with a lower incidence of metabolic syndrome, underscoring the importance of a high-energy breakfast [66].

The influence of eating times on energy balance and metabolism is an active area of research. Human studies have demonstrated that timed nutrient intake may be an efficacious strategy to augment weight loss and improve glycemic control [67]. Mazri FH et al. discovered that lower energy consumption in the early part of the day is associated with increased energy intake later, which elevates the risk of developing unhealthy metabolic states. In alignment with these results, a one-year longitudinal study on adult women reported that higher caloric intake during dinner correlated with poorer diastolic blood pressure outcomes. Furthermore, two randomized controlled trials involving weight loss interventions revealed that participants who consumed the majority of their calories at breakfast (and lunch) and reduced their intake at dinner experienced greater weight loss and a reduction in insulin resistance. A randomized crossover trial indicated that, with equivalent dietary composition, diet-induced thermogenesis (DIT) following dinner was diminished compared to after a morning meal. The attenuation in DIT reflects decreased energy expenditure on the digestion, absorption, and metabolism of nutrients. Consequently, a sustained increase in energy expenditure during the latter part of the day could potentially lead to weight gain [68]. Jeong S et al. utilized a database analysis to demonstrate that even after controlling for age and total energy intake, individuals with higher evening energy intake were more predisposed to develop obesity and metabolic syndrome (MetS) [69].

## 5. Conclusions

Investigations into the correlation between the gut microbiome and health have been conducted; however, the mechanisms governing these interactions remain incompletely understood. Additional research is necessary to elucidate the specific causal mechanisms underlying these relationships. Recognizing the involvement of the gut microbiome in a broad spectrum of diseases, there is considerable interest in pursuing innovative therapeutic strategies. It has been proposed that modulating the composition of the gut microbiome could serve as a preventive or therapeutic measure against certain diseases. Concurrent with the expansion of knowledge regarding the gut microbiome, the potential for personalized medicine is also growing. The future of healthcare may well witness the advent of individualized treatments based on the unique characteristics of a patient’s gut microbiome.

## Figures and Tables

**Figure 1 microorganisms-12-00851-f001:**
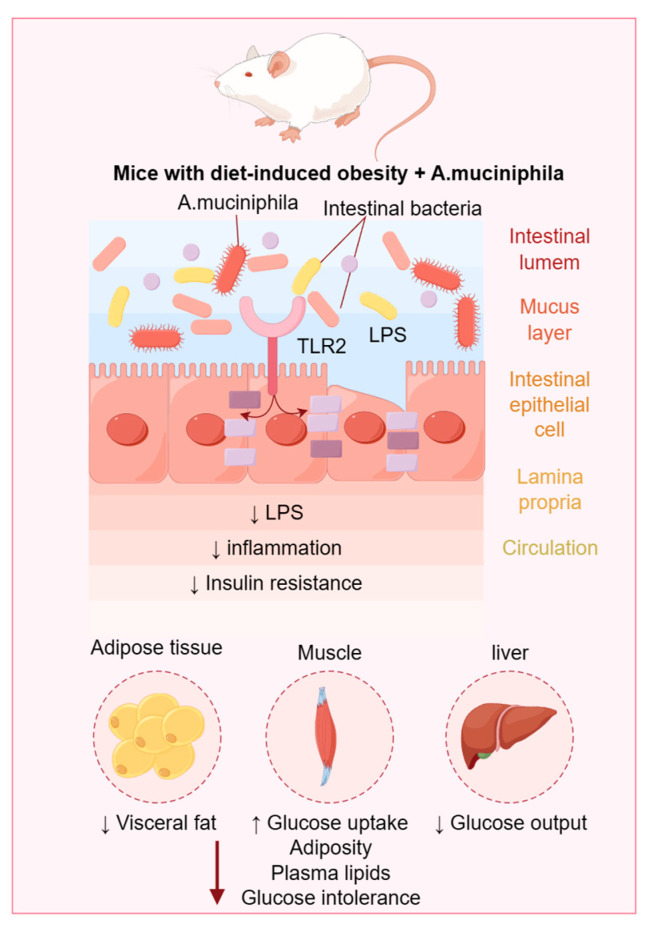
Metabolic syndrome caused by obese mice.

**Figure 2 microorganisms-12-00851-f002:**
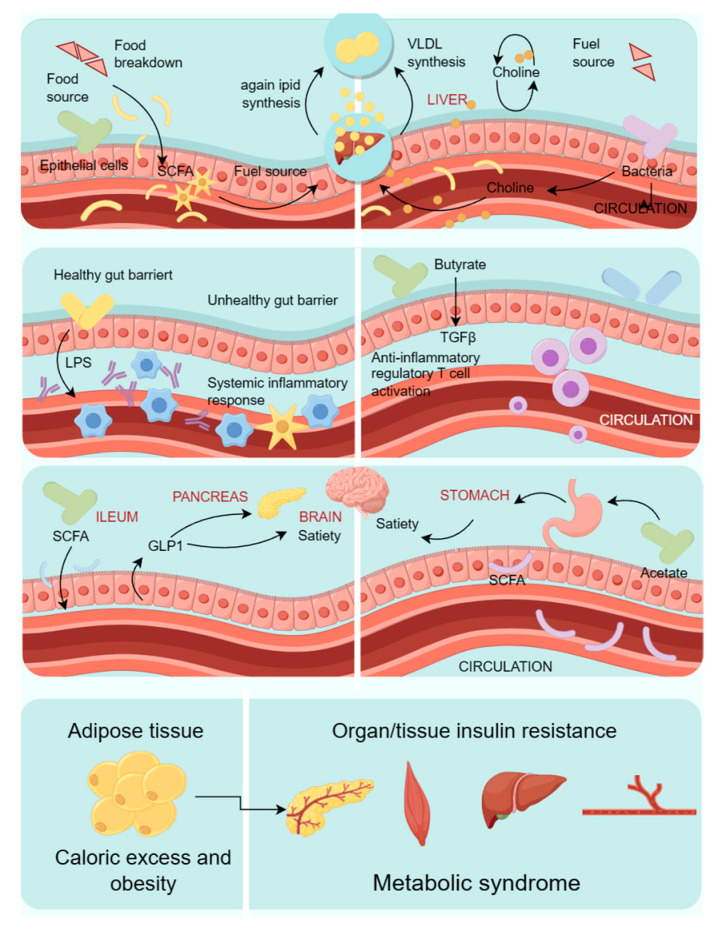
Metabolism of gut microbiota and its effects on target organs.

**Table 1 microorganisms-12-00851-t001:** Registered clinical trials and animal studies about biotherapy to treat metabolic syndrome.

Biotherapy	Resource	Disease	Outcomes	Mechanism of Action	References
**PNS**	Male C57BL/6J mice (About 4 weeks old)	Obesity	PNS reduced adiposity in DIO mice but not in mice with induced obesity and impaired leptin signaling	The leptin-AMPK/STAT3 pathway induced by the PNS-mediated modulations in the gut microbiota was involved in beige adipocyte reconstruction	(Gupta, Osadchiy et al., 2020 [54])
**the “W-LHIT” capsules**	Thirty-seven patients aged 18 to 60 from Wei-En hospital	Obesity	W-LHIT significantly improved body weight and comorbid conditions without obvious adverse reaction or rebound weight gain	Increased abundance of Akkermansia muciniphila and Enterococcus faecium and decreased abundance of Proteobacteria in gut microbiota	(Cao, Wei et al., 2023 [55])
**Probiotics+** **BBR**	T2D patients	Diabetes Mellitus, Type 2	Ant-diabetes effect	BBR can reduce intestinal microbiota bile acid (BA) conversion, thereby reducing intestinal farnesol X receptor (FXR) activity	(Zhang, Gu et al., 2020 [26])
**Intermittent** **Fasting**	Adults with MS, age 30 to 50 years	Cardiometabolic Risk Factors	IF induces a significant alteration of the gut microbial community and functional pathways in a manner closely associated with the mitigation of cardiometabolic risk factors.	IF induced significant changes in gut microbiota communities, increased the production of short-chain fatty acids, and decreased the circulating levels of lipopolysaccharides	(Guo, Luo et al., 2023 [56])
**GP-derived seasonings**	High-risk cardiovascular subjects and healthy subjects	Hyperten sion	GP-seasoning may help in the modulation of cardiometabolic risk factors, mainly in the early stages	Modulation of gut microbiota and functional bacterial communities by grape pomace	(Taladrid, Celis et al., 2022 [42])
**sodium reduction with slow sodium or placebo tablets**	145 participans (42% Black people, 19% Asian, and 34% female)	Hyperten sion	Reducing dietary intake can lower blood pressure and improve arterial compliance	Reducing dietary sodium can increase short-chain fatty acids in the circulation, supporting the potential impact of dietary sodium on human gut microbiota	(Chen, He et al., 2020 [45])

PNS, notoginsenoside R1 and ginsenosides Rb1, Rd, Re, Rf, and Rg1; W-LHIT capsules were prepared in a GMP facility (Tian-jiang Pharmaceutical, Jiangsu, China); BBR, berberine; GP, grape pomace.

## Data Availability

No new data were created or analyzed in this study. Data sharing is not applicable to this article.

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
