# Peer review of "Advances in Gut Microbiota-Targeted Therapeutics for Metabolic Syndrome"

_microorganisms, 2024, doi:10.3390/microorganisms12050851_

Round 1
Reviewer 1 Report
Comments and Suggestions for Authors
The paper presents challenges in readability and comprehension primarily due to its lack of organization, resulting in redundancy throughout the manuscript. While it does not adhere to the systematic review format, even in a narrative review, a coherent structure is essential for understanding the subject matter. To enhance readability, several improvements are recommended:
1. - Streamline the discussion on various pathologies, assuming readers' familiarity with these topics. For instance, sections spanning from line 57 to 80, 110 to 125, and 166 to 179 should be condensed.
2. - Organize the content into subchapters for better clarity, such as:
a. Gut-microbiome pathophysiological evidence and pathology.
b. Diet-gut relationship and its impact on pathology.
c. Effects of microbiome modulation on pathology.
These subchapters should incorporate content from chapters 7, 8, and 9 respectively.
The meaning of the table (which also lacks explanatory footnotes) is obscure. Creating a summary table could enhance the work's citability.
Address superficial editing issues, such as the term "ChronicObesity" (line 100) (I don't think obesity has an acute variant).
Additionally, references are totally disorganized, making it impossible to evaluate the sources.
Reviewer 2 Report
Comments and Suggestions for Authors
The authors reviewed recent papers on the relationship between Gut Microbiota and metabolic syndrome and anticipated future directions in the evolving field. To improve the quality of this paper, the authors should revise it according to the following suggestions;
1) Based on recent research results, "chrononutrition” is attracting attention as a nutritional approach to treating metabolic syndrome. Furthermore, it has been shown that there is a close relationship between gut microbiota and chrononutrition. Nutrition, such as what to eat them, is important in preventing metabolic syndrome, but even more important is thinking about when to eat. At this time them, an item on nutrition should be added.
Round 2
Reviewer 1 Report
Comments and Suggestions for Authors